# Antenatal Care and Health Behavior of Pregnant Women—An Evaluation of the Survey of Neonates in Pomerania

**DOI:** 10.3390/children10040678

**Published:** 2023-04-03

**Authors:** Anja Erika Lange, Janine Mahlo-Nguyen, Guillermo Pierdant, Heike Allenberg, Matthias Heckmann, Till Ittermann

**Affiliations:** 1Department of Neonatology & Paediatric Intensive Care Medicine, University of Greifswald, 17475 Greifswald, Germany; 2Department of Obstetrics and Gynecology, University of Greifswald, 17475 Greifswald, Germany; 3Institute of Community Medicine, Division of Health Care Epidemiology and Community Health, University of Greifswald, 17475 Greifswald, Germany

**Keywords:** antenatal care, health behavior, pregnancy, public health, SNiP study

## Abstract

Background. The German maternity guidelines require regular medical checkup (MC) during pregnancy as a measure of prevention. Socioeconomic factors such as education, profession, income and origin, but also age and parity may influence the preventive and health behavior of pregnant women. The aim was to investigate the influence of these factors on the participation rate in MC of pregnant women. Method. The current analysis is based on the prospective population-based birth cohort study Survey of Neonates in Pomerania, which was conducted in Western Pomerania, Germany. The data of 4092 pregnant women from 2004 to 2008 were analyzed regarding the antenatal care and health behavior. Up to 12 MC were regularly offered; participation in 10 MC is defined as standard screening according to maternity guidelines. Results. Women participated in the first preventive MC on average in the 10th (±3.8 SD) week of pregnancy. 1343 (34.2%) women participated in standard screening and 2039 (51.9%) took a screening above standard. 547 (13.92%) women participated in less than the 10 standard MCs. In addition, about one-third of the pregnancies investigated in this study were unplanned. Bivariate analyses showed an association between better antenatal care behavior and higher maternal age, stabile partnerships and mother born in Germany, *p* < 0.05. On the contrary antenatal care below standard were more often found by women with unplanned pregnancies, less educational women and women with lower equivalent income, *p* < 0.001. Health behaviors also influenced antenatal care. Whereas the risk of antenatal care below standard increased by smoking during pregnancy (RRR 1.64; 95% CI 1.25, 2.14) and alcohol consumption (RRR 1.31; 95% CI 1.01, 1.69), supplementation intake was associated with decreased risk (iodine—RRR 0.66; 95% CI 0.53, 0.81; folic acid—RRR 0.56; 95% CI 0.44, 0.72). The health behavior of pregnant women also differs according to their social status. Higher maternal income was negatively correlated with smoking during pregnancy (OR 0.2; 95% CI 0.15, 0.24), but positively associated with alcohol consumption during pregnancy (OR 1.3; 95% CI 1.15, 1.48) and lower pre-pregnancy BMI (Coef. = 0.083, *p* < 0.001). Lower maternal education was positively correlated with smoking during pregnancy (OR 59.0; 95% CI 28.68, 121.23). Conclusions. Prenatal care according to maternity guidelines is well established with a high participation rate in MC during pregnancy of more than 85%. However, targeted preventive measures may address younger age, socioeconomic status and health-damaging behaviors (smoking, drinking) of the pregnant women because these factors were associated with antenatal care below standard.

## 1. Background

Every day in 2017, 810 women worldwide died from preventable complications during pregnancy or childbirth [1]. In developing countries, only about 65% of women receive prenatal care [2]. In Europe, Australia and the United States binding preventive programs exist that regulate antenatal care [3,4,5,6,7,8]. In Germany, maternity guidelines regulate prenatal care [4]. The aim of these preventive programs is to enable adequate health care for pregnant women and to detect high-risk pregnancies at an early stage [9]. In Germany, ten to twelve preventive medical check-ups are planned for healthy women during pregnancy [9]. The screening programs in other countries show that there are different ideas about the optimal number of precautionary examinations in pregnancy. A comparative survey in nine European countries yielded an average frequency of eight preventive examinations during pregnancy [10]. In the US, the American College of Obstetricians and Gynecologists [4] recommended fourteen examinations as the standard. In 2016, the WHO presented the new antenatal care model (ANC model), according to which at least eight prenatal cares should be established as the standard worldwide [11]. Since the introduction of the maternity guidelines in Germany, an improvement in maternal and perinatal morbidity and mortality has been recorded. The perinatal mortality in Germany before the introduction of the maternity guidelines 1968 was 28/1000 births [BIB], in 2006 it was 5.5/1000 births [12]. The WHO also reported on reduced maternal and perinatal mortality and better pregnancy outcomes worldwide due to the increased use of prenatal care [13,14]. These include the reduction of low birth weight, growth retardation and underweight of the newborn [15].

In addition to socio-demographic factors such as age, parity and relationship status, socio-economic factors such as education, occupation, income and origin, may also affect antenatal care, i.e., the number of prenatal check-ups. According to the European Perinatal Health Report younger women in Europe have a higher risk for a lower social status and for inadequate antenatal care [16]. Furthermore, women with a lower socio-economic status (SES) tend to participate in less antenatal care examinations [17,18,19]. In addition, the health status and health behavior of the pregnant woman also have an impact on the use of preventive medical check-ups. Pregnant women with severe diseases and higher risk for complications should use more antenatal check-ups than women without any risk constellation [9,11,20]. The present study is intended to show the differences in the antenatal care of pregnant women in Western Pomerania depending on socio-demographic and socio-economic influences as well as health and risk behavior.

## 2. Methods

### 2.1. Study Design of SNiP-I

The present study is part of the birth cohort study ‘Survey of Neonates in Pomerania (SNiP-I)’. Data were evaluated with respect to antenatal care and health behavior of pregnant women in Western Pomerania. The SNiP-I study was conducted from February 2002 to November 2008 in the region of Pomerania in Northeastern Germany. SNiP is a population-based, representative study that is able to describe the living and health conditions of newborns and their families comprehensively. The study area was defined by zip code areas 17379, 17389–17999. In addition, newborns from other catchment areas who had been transferred to the Perinatal Center of the University of Greifswald due to premature birth, congenital malformations and small for gestational age were included in the study. In addition to a place of residence outside the above-mentioned zip code areas, a significant language barrier between the parents and the interviewer was an exclusion criterion.

From each participating mother-child dyad, study stuff collected and recorded a total of more than 270 variables on personal data, medical records, and socioeconomic background. An analysis of the data from non-participating pregnant women did not reveal any significant selection bias. The design of SNiP-I study has been described in detail by Ebner et al. [21].

### 2.2. Ethics and Data Protection

The study design was approved by the Ethics Committee of the University Medicine Greifswald (Reg.-Nr III UV 20/00-05/2002). Eligible women were asked for a written informed consent. The study followed the principles of Declaration of Helsiniki. In cases of legally minor mothers the additional signatures of their legal caregivers were required. The collected data was pseudonymized.

### 2.3. Population

The baseline of SNiP-I birth cohort includes data on 5800 mother-child dyads. For purpose of this analysis, the data from the years 2004 to 2008 were evaluated in order to map the data of complete years. This subpopulation from the basline SNiP-I birth cohort comprised 4092 mother-child dyads. 3929 data sets contained information on antenatal care (Figure 1).

### 2.4. Maternity Guidelines

In Germany, the maternity guidelines regulate which prenatal check-ups should be carried out in which period [9]. In addition to physical and serological examinations, three ultrasound screenings and counseling sessions on health-related topics are included. All examinations are documented in the maternity record. In the maternity directives in Germany, the necessary examinations during pregnancy covered by the health insurance companies. In our study, we have limited ourselves exclusively to preventive examinations that are also covered by health insurance companies in order to curb the financial effect on the implementation of the examination.

### 2.5. Antenatal Care

The time of the first check-up and the number of all obstetric check-ups in pregnancy were recorded from the maternity record. Based on other national studies [22,23,24,25] and on the basis of the maternity guidelines [9], ten to twelve medical checkups are considered *standard*. Participation in less than ten examinations was classified as *antenatal care below standard*. Participation in more than twelve examinations for uncomplicated pregnancies was classified as *antenatal care above standard*. Based on the considerations of the WHO [26], fewer than five medical check-ups were considered to be *inadequate care* and fewer than one or no medical check-up as *completely inadequate care* [26,27].

### 2.6. Potentially Predictive Variables of Antenatal Care

#### 2.6.1. Socio-Demographic Factors

The age of the women was queried in full years. According to the maternity guidelines, women under the age of 18 and older than 35 are considered a risk group [9]. Accordingly, women were divided into three categories: <18 years, 18 to 35 years and >35 years. Parity describes the number of children born to a woman. Accordingly, pregnant women were classified as primipara, secundipara and multipara. Partnership was classified as *living in a stable partnership* or *single.* Origin was classified in *German* or women *with a migration background.*

#### 2.6.2. Socio-Economic Factors

Socio-economic factors included educational level, professional status and income of pregnant women and the partner/huesband. The stratification pattern for educational level followed previously published patterns [28,29]. Persons being still at school, without school diploma, or with five years or less of secondary school, were grouped as having a lower educational level. Persons with six years of secondary school (German ‘Realschulabschluss’) were included in second level—middle educational level. The third level—high educational level—included persons with eight years of secondary school (‘German Fachhochschulreife’ or ‘Abitur’). The last level included persons with university degree. Professional status was classified into four categories: A. self-employed, B. employed, C. in education and D. unskilled. Income was asked in the form of a household net income. Twelve income categories were available for this in the self-answer questionnaire.

#### 2.6.3. Health Behavior

The concept of health behavior contains both health-damaging behaviors and health-promoting behaviors. In the health-damaging behavior category, information on pre-pregnancy body mass index (BMI), smoking and alcohol consumption during pregnancy were evaluated. For BMI calculation we used height (in cm) and pre-pregnancy body mass (in kg) which were reported by women using a standardized self-report questionnaire. According to the classification recommended by WHO [30] women were categorized into different BMI groups. Following previously published patterns [28,29], this study used a dichotomous classification for the information on smoking and alcohol consumption: ‘smoker/non-smoker’ and ‘drinker/non-drinker’. A pregnant woman was classified as a smoker if she declared to smoke during the last four weeks before delivery. Similarly, a woman was classified as a drinker if she declares to continue to drink alcohol during pregnancy, irrespective of time period or amount of consumption. To record alcohol consumption, questions from the AUDIT-C [31], a short questionnaire to identify problematic alcohol consumption, was used. Health-promoting behaviors included the intake of supplements such as iodine, folic acid, iron and magnesium. Supplementation intake was classified dichotomous in ‘intake—yes’ or ‘intake—no’.

#### 2.6.4. Maternal Factors

Maternal factors that were analyzed were planned pregnancies and general maternal conditions or chronic diseases for which the pregnant woman had to take medication (yes or no). Pregnancy planning was classified in ‘planned pregnancy’, ‘no pregnancy planning, but no use of contraception’ and ‘pregnancy despite using contraception’.

### 2.7. Statistical Analysis Strategies and Test Methods

The collected data was stored in a Microsoft Access database. After a corresponding data usage request to the study management, the statistical data analysis was performed using STATA for Windows. Means and SD were used to describe metric variables and percentages and frequencies to describe categorical variables. In addition to the descriptive analysis of the sample, relative risk ratios and odds ratios were used to describe the risk of the different exposures. Associations between socio-demographic, socio-economic, maternal factors, and health behavior with antenatal care behavior were calculated using a multinominal regression model. For the multinominal logistic regression model we divided the antenatal care into three groups according into low, high, and normal antenatal care. The cut-offs for this were derived from guidelines (cut offs ≤9 and ≥12). In our analyses we aimed to find markers for low and high antenatal care. For this, the multinomial logistic regression is the model of choice because it can handle categorical outcomes with more than two levels. We did a Small-Hsiao- test to test the assumption of irrelevant alternatives. This showed that the assumption was not violated in our model (*p* = 0.864).

## 3. Results

### 3.1. Antenatal Care Behavior of the Studied Subpopulation

Out of 4092 mother-child dyads participating in the SNiP-I baseline study from 2004 to 2008, antenatal care behavior in pregnancy was reported for 3929 mother-child dyads. The mean number of antenatal MCs was 12.8 ± 3.5. On average, the first MC took place at the 10th (mean ± 3.8 SD) week of pregnancy.

Overall, 1343 mothers had standard antenatal care (34.2%) and 2039 had antenatal care above standard (51.9%). 547 mothers (13.9%) had an antenatal care below standard. In detail, 11 women of them (0.28%) did not attended any or at most one screening (*totally inadequate antenatal care*). 61 (1.55%) of them attended two to five (*inadequate antenatal care*) and 475 (12.09%) attended six to nine screening examinations.

### 3.2. Maternal and Pregnancy Characteristics of the Studied Subpopulation

The women (n = 4092) interviewed in the SNiP study from 2004 to 2008 had an average age of 27.5 ± SD 5.4 years. 3974 (97.3%) of them were born in Germany. 3308 (93.3%) of the women lived in a steady relationship, 1244 (37.9%) of them were married. Parity information could be evaluated from 3928 women. 1803 (45.9%) women had their first child, for 2125 (54.1%) women it was already at least their second child.

Data on pregnancy planning were available from 3367 (82.3%) pregnant women. 2308 (68.6%) stated that the pregnancy had been planned. Of the remaining women (n = 1059, 31.5%), 722 (21.4%) stated that they had not used contraception and 337 (10.0%) stated that they had become pregnant despite using contraception. Information on chronic diseases was provided by 3775 (92.3%) pregnant women. About one third of the women had at least one such disease during pregnancy (n = 1327; 35.2%).

Maternal and pregnancy characteristics stratified by the participation in MCs during pregnancy in the study population are presented in Table 1. Bivariate analyses showed that younger women, those who had a shorter gestational time until birth and those who were not born in Germany had antenatal care below standard. Women with unplanned pregnancy had also more often antenatal care below standard than the mothers in the other two groups. Mothers with antenatal care above standard did not differ substantially compared to mothers with standard antenatal care. The most profound differences between these two groups were observed for parity and maternal chronic diseases. Mothers with chronic diseases had more MCs compared to mothers without chronic diseases. Higher parity seems to be associated with standard antenatal care compared to antenatal care above standard.

### 3.3. Maternal Socio-Economic Characteristics of the Studied Subpopulation

Around 70% of the women surveyed from 2004 to 2008 (N = 4092) provided information about their socioeconomic status. There were data on education level from 3534 (86.4%) pregnant women in the survey. 550 (15.6%) women could be assigned to a low educational level, 1824 (51.6%) to a middle educational level and 590 (16.7%) to a high educational level. 570 (16.1%) of the pregnant respondents had a university degree. The professional status could be evaluated from a total of 2919 (71.3%) women. 127 (4.3%) of them stated that they were self-employed and 2269 (77.8%) were employed. In addition, 206 (7.1%) pregnant women were still in training and 317 (10.9%) were unskilled. The mean income in the studied population was 1146€ (± SD 680€), *p* < 0.001.

Analyses showed that less educational women and women with lower equivalent income more often had antenatal care below standard, *p* < 0.001. The maternal socio-economic characteristics in relation to participation in medical check-up during pregnancy are shown in Table 2.

### 3.4. Maternal Health Behavior Characteristics of the Studied Subpopulation

#### 3.4.1. Health-Damaging Behavior

There was a response rate of around 85% regarding health-damaging behavior. There were 644 (18.5%) smokers in the study population. 913 (25.5%) respondents reported having consumed alcohol during their pregnancy. Analyses showed that higher maternal income was negatively correlated with smoking during pregnancy (OR 0.2; 95% CI 0.15, 0.24), and positively associated with alcohol consumption during pregnancy (OR 1.3; 95% CI 1.15, 1.48). Besides women with higher income had on average a lower BMI (Coef. = 0.083, *p* < 0.001). Lower maternal education, on the other hand, was positively correlated with smoking during pregnancy (OR 59.0; 95% CI 28.68, 121.23).

#### 3.4.2. Health-Promoting Behavior

Data on supplement intake during pregnancy were available from all 4092 pregnant women. Table 3 shows the supplement intake. Overall, more than half of the pregnant women took supplements (iodine: n = 2674, 65.4%; folic acid: n = 3291, 80.4%; iron: n = 2359, 57.7%, magnesium: n = 1906, 46.6%). On average, women began taking folic acid in the 2nd month of pregnancy (1.96 month ± SD 1.4). Supplement intake was positively associated with maternal education (*p* < 0.005).

#### 3.4.3. Health Behavior and Antenatal Care

Selected health behavior in relation to participation in medical check-up during pregnancy are shown in Table 3. Smoking during pregnancy, alcohol consumption, lower supplementation intake of folic acid, iron, magnesium and iodine were more often found by mothers with antenatal care below standard compared to the groups with more than 9 visits. Mothers with antenatal care above standard did not differ substantially compared to mothers with standard antenatal care. The most profound differences between these two groups were observed for smoking during pregnancy.

### 3.5. Association of Selected Variables with Antenatal Care Behavior Adjusted for Gestational Age at Birth

Table 4 show associations between socio-demographic, socio-economic, maternal factors, and health behavior with antenatal care beh\avior (*p* < 0.05). The associations were calculated using a multinominal regression model adjusted for gestational age at birth.

#### 3.5.1. Socio-Demographic Factors Influencing Antenatal Care Behavior

Maternal age, living with a partner and mother born in Germany were associated with a better antenatal care behavior (*p* < 0.05). The risk of antenatal care below standard decreases by 5% per additional year of maternal age. Permanent partnerships of pregnant women also reduce the risk of antenatal care below standard by 34%. Mothers born in Germany had a 49% risk decrease for antenatal care below standard.

#### 3.5.2. Socio-Economic Factors Influencing Antenatal Care Behavior

The risk of having antenatal care below standard decreased by 41% per 1000€ of household income.

#### 3.5.3. Maternal Factors Influencing Antenatal Care Behavior

If the pregnancy was not planned, the risk of antenatal care below standard increases by 74% compared to a planned pregnancy. Women with chronic diseases attended more MCs.

#### 3.5.4. Health Behavior Factors Influencing Antenatal Care Behavior

Health-damaging behaviors (smoking, drinking) were associated with antenatal care below standard. In contrast, taking dietary supplements (intake of iodine, folic acid, and magnesium) had a positive effect on the participation in MCs.

### 3.6. Impact of Selected Maternal, Pregnancy and Health Behavior Factors on Antenatal Care Behavior

We calculated a multiple logistic regression model to evaluate the association (all variables with *p* < 0.05) between antenatal care and each of the studied maternal, pregnancy and health behavior factors (gestational week of birth, parity, educational level, pregnancy planning, pre-pregnancy BMI, supplementation of magnesium) (OR 95% CI; Table 5). The model demonstrated a modest fit (AUC = 0.787). If maternal, pregnancy and health behavior factors were viewed as cofounders, the likelihood of participating in standard antenatal care increased among the gestational week of pregnancy, among women with at least middle educational level and among the willingness of supplementation intake. The likelihood of antenatal care below standard increased by 1.59 respectively 2.58 by unplanned pregnancies and by 1.52 if its not the first child of pregnant women.

### 3.7. Discussion

The aim of this study was to analyze the antenatal care behavior and health behavior of pregnant women and investigate the influence of sociodemographic and socioeconomic factors on these behaviors. The analyses showed that the social status of the mother can have a major impact on her participation in antenatal care and her health behaviors.

#### 3.7.1. Antenatal Care Behavior

In the study region, antenatal care according to maternity guidelines is well established with a high participation rate in standard or above standard MC during pregnancy of 86% and an overall participation rate about 98%. However, a relevant number of pregnant women received antenatal care below standard, i.e., participated in less than the 10 MCs. It is worth noting that the 2016 WHO antenatal care guidelines recommend a minimum of eight antenatal care visits to reduce perinatal mortality and improve women’s experience of care [13]. This goal was found to be met by the majority in our survey. However, internationally, this is still a different story. An analysis of antenatal care in low-income country middle-income countries showed that participation in at least four preventive examinations must already be considered a success, but this is nevertheless not achieved for the most part [32].

Compared with international studies, the fact that the majority of the women we studied attend their first screening mostly around the 10th week of pregnancy is a good indicator of careful antenatal care, especially because the WHO [33] provides first antenatal screening by the 12th week of pregnancy. Internationally, it was shown that in 2013, the participation rate in early antenatal care in developing regions was 48.1% compared to 84.8% in developed regions [34]. In turn, a later start of antenatal care can lead to poorer fetal outcomes such as low birth weight and prematurity [35]. Again, the level of education as well as the socioeconomic status of the mother were found to be the most important determinants [36].

#### 3.7.2. Risk Factors for Antenatal Care Below Standard

Preterm births contributed significantly to antenatal care below standard in our study. However, our data do not allow to differentiate whether preterm birth occurred despite regular attendance to MCs or because of insufficient attendance to MCs. Further investigation should be done to differentiate.

After controlling for preterm birth sociodemographic and socioeconomic factors were identified to be associated with antenatal care below standard. Women attending antenatal care below standard were for example on average three years younger and also had lower incomes and lower education than pregnant women with at least standard antenatal care. This is congruent with data from the European Perinatal Health Report, which showed that younger women in Europe are at higher risk for lower social status and inadequate antenatal care [16]. Our findings are also consistent with other previous national and international studies that showed an association between inadequate antenatal care and low maternal age [37], low income [38], and low educational level [39]. As well, (single) women of non-German origin were previously identified as a risk group for antenatal care below standard [23].

Origin was associated with antenatal care below standard also in our study. Although the proportion of non-German participants in our study population was very low this is an important issue. Moreover, other national surveys found similar results with higher rates of participation among non-German pregnant women [23]. Racial and ethnic disparities considerably contribute to maternal morbidity [40,41].

Around one-third of the pregnant women we surveyed said that their pregnancies had been unplanned or had occurred despite contraception. This was another risk factor for antenatal care below standard in our study. However, our data do not allow to represent whether the unplanned pregnancy was detected later, leaving less time for adequate prenatal care, or whether pregnant women deliberately avoided adequate prenatal care. This and the question of how such a high number of unplanned pregnancies could occur should be the subject of further investigation.

Nevertheless, our results of antenatal care below standard for unplanned pregnancies are consistent with international findings, that have demonstrated that unplanned pregnancies were directly associated with poor health care utilization [42], late initiation of antenatal care and inadequate use of antenatal screenings [43].

This unequal distribution in antenatal care illustrates that despite freely accessible medical care and a standardized range of care through maternity guidelines [9], equal opportunities are not yet equally guaranteed for all women. In order to prevent a lack of prevention and its consequences for mother and child, prevention work and good education by practicing gynecologists in these sub-communities is of enormous importance even before pregnancy begins [44]. Here, consideration should be given to whether the health insurance funds should not work more intensively with bonus programs for preventive measures that maintain and promote health and thus create an additional incentive for targeted preventive health care [45].

#### 3.7.3. Health Care Behavior

Health-damaging behaviors, such as smoking, have been shown to be more common among people of low socioeconomic status [46]. These differences contribute to health disparities during pregnancy [47]. In the SNiP cohort, the incidence of smoking during pregnancy was approximately 20%. Moreover, this adverse health behavior was associated with inadequate antenatal care and lower socio-economic status. Maternal smoking during pregnancy is one of the most important modifiable determinants of low birth weight and other adverse perinatal outcomes [48]. These results are both higher than smoking rates reported in other national studies [49] than those reported in other European countries. For example, smoking rates range from less than 5% in Sweden to 17% in France or 19% in Scotland [16].

Therefore, specific antenatal counseling for women who smoke is needed as part of antenatal care. The importance of smoking cessation in early pregnancy was recently underscored by the finding that no major growth deficits were observed in women who quit smoking in early pregnancy [50] (Lit). However, it was also found that only one in four women quit smoking during pregnancy [51]. In addition, antenatal care should include counseling on lifestyle-related stresses for the father and the entire family [52].

While smoking was associated with low socioeconomic status in the SNiP study, alcohol use was more prevalent among women in higher income brackets.

Although alcohol consumption is known to cause preterm delivery, stillbirth [53], fetal growth retardation, or fetal alcohol spectrum disorders [54] our results showed that nearly one-quarter of women continued to consume alcohol during pregnancy. It should be noted, however, that alcohol consumption in general during pregnancy was collected and evaluated in our study, regardless of timing and amount of alcohol consumption. However, compared with U.S. surveys [55] in which nearly half of women drank alcohol during pregnancy, our results are significantly lower. Compared with an international multicenter study [56], the results of our analyses are also in the lower range. Thus, alcohol consumption during pregnancy was found to vary widely both within and between countries. Among others, alcohol consumption was examined in the United Kingdom, Australia and New Zealand and varied between 40% and 80%. This may be because until a few years ago, recommendations in the UK assumed that low levels of alcohol consumption were unlikely to be harmful [5]. However, current guidelines refute this, as there is insufficient evidence to support the safety of any alcohol consumption during pregnancy [57]. Accordingly, any type of alcohol consumption during pregnancy should be viewed critically, especially since fetal alcohol spectrum disorder is a consequence for which there is currently no treatment or established diagnostic or therapeutic tools to prevent and/or reduce the associated adverse consequences [54]. Therefore, preventive approaches should be used in the pre-conception and pre-birth periods to intervene in a timely manner, mitigate the effects of fetal alcohol spectrum disorder, and ideally prevent it for life. In addition, it is important to remember that alcohol use is nevertheless stigmatized in society. Accordingly, it is unclear whether the reported values correspond to actual consumption or whether there may not be an even higher number of unreported cases.

In contrast, health prevention behavior, i.e., the intake of folic acid, iodine and magnesium was associated with at least standard antenatal care. Furthermore, we found that more educated pregnant women were more likely to take supplements than pregnant women with a low level of education. This is relevant insofar as the socioeconomic status again can play a decisive role with regard to the child’s health. For this reason, the World Health Organization have also developed nutritional recommendations and strategies to prevent adverse pregnancy outcomes [13]. Accordingly, more effective education of all women of childbearing age would be needed here as well, because maternal nutrition plays a key role in fetal development and neonatal growth [58]. Although 80% of women in our survey were taking folic acid, on average they also did not start until the second month of pregnancy. However, our data do not allow a more precise differentiation of the reasons. Further studies proved not only a low level of education, but also unplanned pregnancies or pregnancies that occurred earlier than expected as a risk factor. Here, the women concerned stated that they no longer had enough time for the correct start of intake. Furthermore, a lack of information as well as the conscious decision not to take the medication was highlighted as a risk factor for insufficient supplementation [59]. Furthermore, another factor to be investigated could be that the cost of supplementation is usually borne by the pregnant woman herself. For women with a low level of education and thus potentially low income, this could be an additional burden that they might not be willing to bear if they are not sufficiently informed about the consequences of non-adherence.

#### 3.7.4. Strength and Limitations

The strengths of our analysis are the high population coverage of SNiP-I, the large number of participants, geographically defined study region, homogeneous ethnic compositions and a comprehensive dataset including medical and socio-economic factors. A limitation of the study was that we used health behavior data from self-reported questionnaires of pregnant women.

A limitation of the study was that we used the health behavior regarding alcohol and tobacco consumption from the mothers’ self-reports, that could be a a source of error. It would be possible that the women have answered in terms of socially desirable behavior. Accordingly, the actual number of women smokers and alcohol consumers could be higher. Nevertheless, our prevalence for alcohol consumption and smoking [60] was already higher than that reported in other studies for Germany [61]. Moreover, we calculated the pre-pregnancy BMI using mothers’ self-reported data of height and weight, which might be also a source of error. Mental decision-making is an extremely important factor. Unfortunately, a detailed psychiatric examination could not be carried out on the large number of patients, but the known psychiatric illnesses and medications taken were inquired about and documented from the patient files or maternity records.

## 4. Conclusions

Antenatal care according to maternity guidelines is well established, and the participation rate in MC during pregnancy is high, above 85%. Nevertheless, younger age, lower socioeconomic status and low educational level were identified as risk factors for antenatal care below standard. In addition, these risk factors influence health behaviors, which in turn influence women’s antenatal care. Accordingly, targeted preventive measures should be aimed at these risk groups as well as health-damaging behaviors (smoking, drinking) of pregnant women to ensure adequate care for mother and child during pregnancy.

## Figures and Tables

**Figure 1 children-10-00678-f001:**
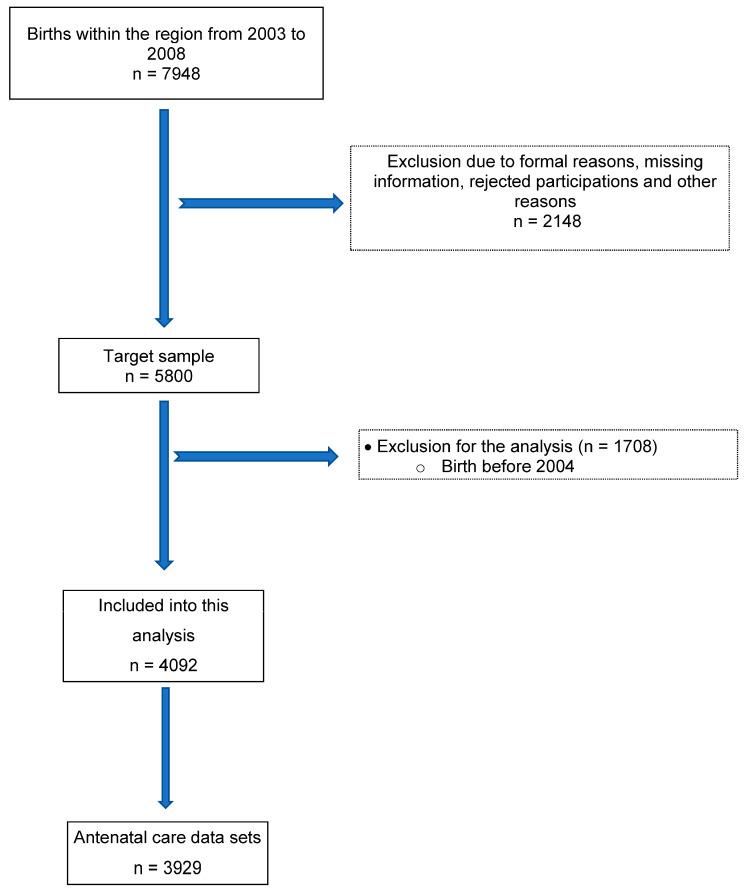
Flow diagram showing selection procedure apply to data from SNiP-I Study (Survey of Neonates in Pomerania).

**Table 1 children-10-00678-t001:** Maternal and pregnancy characteristics stratified by the participation in medical check-up during pregnancy in the SNiP-I cohort from 2004 to 2008.

Variable	N	Antenatal Care			*p*-Value
		≤9 Visits (n = 547)	10–12 Visits (n = 1343)	>12 Visits (n = 2039)	
Age of mother; years	3929	26.8 (5.8)	27.8 (5.3)	27.6 (5.3)	0.0055
Mother born in Germany	3924	96.0%	97.7%	97.4%	0.113
Living with a partner	3426	90.5%	93.6%	93.2%	0.082
Parity 1st child 2nd child 3rd child 4th child or more	3928	41.3% 32.5% 12.3% 13.9%	43.6% 31.2% 15.5% 9.8%	48.7% 29.3% 11.8% 10.2%	<0.001
Maternal chronic disease	3775	30.7%	33.9%	37.2%	0.011
Number of antenatal care visits	3929	7.5 (2.0)	11.1 (0.8)	15.4 (2.6)	<0.001
Gestational week of birth <32 weeks 32–36 weeks 37–41 weeks >41 weeks	3929	37.0 (3.3) 7.8% 20.1% 71.3% 0.7%	38.7 (1.7) 0.7% 6.5% 92.3% 0.6%	39.4 (1.6) 0.4% 3.7% 94.3% 1.6%	<0.001
Planned pregnancy Yes No contraception No	3255	56.2% 27.7% 16.1%	70.0% 21.5% 8.5%	70.8% 20.2% 9.0%	<0.001

N = total number of mother-child dyads available for analyses. Data are expressed as mean and standard deviation (continuous data) or as percentage (categorical data).

**Table 2 children-10-00678-t002:** Maternal socio-economic characteristics in relation to participation in medical check-up during pregnancy in the SNiP-I cohort from 2004 to 2008.

Variable	N	Antenatal Care			*p*-Value
		≤9 Visits (n = 547)	10–12 Visits (n = 1343)	>12 Visits (n = 2039)	
Education of the mother <10 years 10 years 11–13 years University degree	3409	26.1% 44.4% 15.9% 13.7%	12.8% 53.8% 16.8% 16.6%	14.4% 52.1% 16.9% 16.6%	<0.001
Equivalent income; 1000€	2428	1.00 (0.64)	1.2 (0.7)	1.1 (0.7)	<0.001

N = total number of mother-child dyads available for analyses. Data are expressed as mean and standard deviation (continuous data) or as percentage (categorical data).

**Table 3 children-10-00678-t003:** Selected health behavior in relation to participation in medical check-up during pregnancy in the SNiP-I cohort from 2004 to 2008.

Variable	N	Antenatal Care			*p*-Value
		≤9 Visits (n = 547)	10–12 Visits (n = 1343)	>12 Visits (n = 2039)	
BMI before pregnancy; kg/m^2^	3452	23.1 (4.9)	23.5 (4.6)	23.8 (4.9)	0.140
Smoking during pregnancy	3351	27.3%	19.3%	15.5%	<0.001
Alcohol during pregnancy	3455	27.1%	24.5%	26.0%	0.476
Folic acid intake in pregnancy	3929	71.9%	81.5%	82.2%	<0.001
Iron intake in pregnancy	3929	54.5%	59.0%	58.6%	<0.001
Magnesium intake in pregnancy	3929	38.6%	44.1%	51.0%	<0.001
Iodine intake in pregnancy	3929	56.1%	66.5%	67.7%	<0.001

N = total number of mother-child dyads available for analyses. Data are expressed as mean and standard deviation (continuous data) or as percentage.

**Table 4 children-10-00678-t004:** Association of selected variables with antenatal care behavior adjusted for gestational age at birth.

	Antenatal Care below Standard (≤9) vs. Standard Antenatal Care (10–12)	Antenatal Care above Standard (>12) vs. Standard Antenatal Care (10–12)
	RRR (95%-CI)	RRR (95%-CI)
**Maternal age; years**	0.95 (0.94; 0.98) *	1.00 (0.98; 1.01)
**Parity** **2nd vs. 1st child** **3^rd^ vs. 1st child** **4^th^ or more vs. 1st child**	1.21 (0.95; 1.54) 0.90 (0.65; 1.25) 1.53 (1.09; 2.14) *	0.83 (0.70; 0.98) 0.70 (0.56; 0.87) 0.97 (0.76; 1.24)
**Living with a partner**	0.66 (0.44; 0.99) *	0.93 (0.69; 1.26)
**Mother born in Germany**	0.51 (0.29; 0.90) *	0.92 (0.59; 1.46)
**Educational status** **10 years vs. <10 years** **>10 years vs. <10 years** **University degree vs. <10 years**	0.39 (0.29; 0.53) * 0.41 (0.28; 0.60) * 0.37 (0.25; 0.55) *	0.86 (0.68; 1.08) 0.92 (0.70; 1.21) 0.89 (0.68; 1.17)
**Income; 1000€**	0.59 (0.48; 0.74) *	0.96 (0.84; 1.09)
**Planned pregnancy** **No, but no use of contraception vs. yes** **No, but pregnancy despite contraception** **vs. yes**	1.74 (1.32; 2.29) * 2.85 (2.00; 4.06) *	0.91 (0.75; 1.10) 1.00 (0.76; 1.32)
**Maternal chronic diseases**	0.76 (0.60; 0.96) *	1.21 (1.04; 1.41) *
**Pre-pregnancy BMI**	0.98 (0.96; 1.01)	1.01 (0.99; 1.03)
**Smoking during pregnancy**	1.64 (1.25; 2.14) *	0.78 (0.64; 0.95) *
**Alcohol consumption during pregnancy**	1.31 (1.01; 1.69) *	1.00 (0.84; 1.19)
**Intake of jodine during pregnancy**	0.66 (0.53; 0.81) *	1.04 (0.90; 1.21)
**Intake of folic acid during pregnancy**	0.56 (0.44; 0.72) *	1.09 (0.91; 1.31)
**Intake of iron during pregnancy** **Intake of magnesium during pregnancy**	0.86 (0.70; 1.06) 0.59 (0.47; 0.74) *	0.98 (0.85; 1.13) 1.54 (1.33; 1.77) *

Note. Relative risk ratios (RRR) and 95% confidence intervals (CI) were calculated using multinomial regression models adjusted for the gestational week at birth. * *p* < 0.05.

**Table 5 children-10-00678-t005:** Impact of selected maternal, pregnancy and health behavior factors on antenatal care behavior: multiple logistic regression analysis with antenatal care below standard as the dependent variable.

	OR	95% CI		*p*
**Gestational week of birth**	0.64	0.61	0.68	0.000 ***
**Parity** **Second child**	1.52	1.18	1.96	0.001 **
**Pregnancy planning** **No, but no use of contraception** **No, but pregnancy despite contraception**	1.59 2.58	1.19 1.800	2.11 3.68	0.002 ** 0.000 ***
**Educational level** **middle educational level** **High educational level** **University degree**	0.44 0.42 0.44	0.32 0.28 0.29	0.60 0.63 0.66	0.000 *** 0.000 *** 0.000 ***
**Pre-pregnancy BMI**	0.97	0.95	1.00	0.030 *
**Supplementation of magnesium**	0.52	0.41	0.67	0.000 ***

n = 2882 pregnant women (70.4% of the 4092 included in this analysis). *** *p* < 0.001, ** *p* < 0.001, * *p* < 0.05.

## Data Availability

The data of the SNIP-study is publicly available. This is a data repository where any researcher can register and find data dictionary as well as an online application tool for getting access to data. Upon an application by registered users, the Research Cooperation Community Medicine (RCC) of the University of Greifswald, Germany, which is funded by the Federal Ministry of Education and Research (grant no. ZZ 96030) decides on granting access to the data based on scientific guidelines.

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
