# Peer review of "Antenatal Care and Health Behavior of Pregnant Women—An Evaluation of the Survey of Neonates in Pomerania"

_children, 2023, doi:10.3390/children10040678_

Round 1

Reviewer 1 Report

Thank you very much for sharing this article. I have several minor comments.

1. The authors used multinomial logit models. Can you briefly discuss the assumption on the independence of irrelevant alternatives? Did you test this?

2. Would you recommend useful policy suggestions based on study findings?

3. Perhaps the authors can further discuss others types of limitations such as cross-sectional design of the survey, selection bias, and not including partner's characteristcs etc. 

Reviewer 2 Report

This manuscript describes a cohort study in which 4092 pregnant women from Pomerania region were enrolled for the antenatal care behavior analysis. The authors collected the social-demographic data and socio-economic data, and found the relationship between the antenatal care behavior and maternal age, partnership status and children planning of the pregnant women. This topic of antenatal care choice of pregnant women is rarely touched, and the abundant data and large sample size are decent, all of which brings this study novelty and scientific significance. I listed several major concerns need to be addressed. 

1.    The childbearing, and regular medical checkup are more a family plan rather than a personal choice. The study subjects should be families or couples, instead of expectant mothers alone. The pregnant women commonly rely on their spouses or parents as to decision-making. Did the authors investigate the religion status and tradition profile of the originated families or newly coupled families where the pregnant women come from?

2.    What is the financial arrangement of the antenatal care in the local healthcare system? Whether or not these services are covered by public finance or partially charged to the patients, will influence the choice of the pregnant women.

3.    What can highly affect the pregnant women’s decision-making is their own mood/mental status. Did the authors check out the patients’ mood/ psychiatric profile?

Round 2

Reviewer 2 Report

This manuscript describes a cohort study in which 4092 pregnant women from Pomerania region were enrolled for the antenatal care behavior analysis. The authors collected the social-demographic data and socio-economic data, and found the relationship between the antenatal care behavior and maternal age, partnership status and children planning of the pregnant women. This topic of antenatal care choice of pregnant women is rarely touched, and the abundant data and large sample size are decent, all of which brings this study novelty and scientific significance. The authors responded well to my questions and made some revisions. After explanations, the conclusion is less vulnerable. Please put your points of your response to me into the main text, especially limitation part, to feed the readers. 

Author Response

Vielen Dank für Ihr positives Feedback. Ich habe die entsprechenden Punkte wie gewünscht in das Manuskript integriert und den Methodenteil entsprechend ergänzt und die Nichterfassung des psychiatrischen Zustands in den Grenzen der Studie ergänzt.